# The Impact of Stress on Life, Working, and Management Styles: How to Make an Organization Healthier?

**Ivana Katić [1], Tatjana Knežević [1], Nemanja Berber [2] , Andrea Ivanišević [1,\*] and Marjan Leber [3]**

[1] Faculty of Technical Sciences, University of Novi Sad, Trg Dositeja Obradovića 6, 21101 Novi Sad, Serbia
[2] Faculty of Economics in Subotica, University of Novi Sad, Segedinski put 9-11, 24000 Subotica, Serbia
[3] Faculty of Mechanical Engineering, University of Maribor, 2000 Maribor, Slovenia
\* Correspondence: andreai@uns.ac.rs

**Abstract:** This article provides guidelines for optimizing organizational management styles and achieving a balance between life and work. Contributing to sustainable human development will contribute to the psychology of sustainability and sustainable development. The main purpose of the paper is to determine the relations between the preferences of management styles, working styles and lifestyles, and exposure to stress in the managerial population in order to achieve harmonization. A correlation study was conducted on a sample of 618 subjects using the Blanchard test of situational leadership, the Four-Dimensional Symptom Questionnaire for determining the stress levels, the modified Allport–Vernon–Lindsay Scale of Values, and Julie Hay's Working Styles Questionnaire. The paper provides insight into the contribution of management styles to the balance of private and professional areas of life, as well as to stress reduction in managers.

**Keywords:** employee health; organizational health; minimizing stress; lifestyles; working styles; management styles; sustainable development

## 1. Introduction

The need for development in terms of the health of employees and organizations is more pronounced than ever before. Achievements in the field of science about sustainability and the emergence of the psychology of sustainability and sustainable development have contributed to the collection of evidence about modern organizations needing to work in a healthy way and nurture healthy and motivated employees [1].

In their work managers prefer working and management styles that are conditioned by their personality structure. The psychology of sustainability and sustainable development can improve interpersonal relationships in the organization as well as in the environment overall. The preference of the working style and lifestyle of managers is significant from the perspective of integrating the individual into the organization. The problems that are related to the way of managing the organization, the existence of stress, and how employees fit into the lives of their professional choices become a priority, because the existing concept is untenable. The question of whether the problem exists is no longer raised, but rather how many active organizations and managers want to participate in its resolution? A special contribution of this paper is its holistic approach, the observation of all phenomena together. The research identifies weak points in an organization in dealing with the existence of stress, inadequate managerial, working and lifestyles for employees, with a strong focus on maintaining health and optimal work [2].

The authors especially want to draw attention to the awareness that organizations have implemented measures to improve the health of employees, because without healthy employees there

is no healthy and successful organization. Raising awareness about the importance of managers can be promoted through training in the direction of adopting soft skills such as stress management, time management, delegation, etc. Managers with better knowledge and tools in the workplace will be satisfied and the sustainability of human resource practices will have an impact on the organization [3].

The scope of our research includes an analysis of the relationship between the effective management of the organization and the professional style or lifestyle of the manager. It was necessary to investigate the extent to which a manager's stress is present. The survey was seen both from the aspect of the organization and from that of the employees. For research purposes, there is a contribution to identifying risks in business and contributing to sustainable welfare for managers and organizations in general. This study strives to point out the gap in the relationship between organizational values and the styles of a manager's work and ways of reducing that gap [4].

The main aim of this paper is to examine the relationships between the preferences of certain styles of management, working and lifestyles, and exposure to stress in the managerial population. Hence, a hypothesis is set up: We expect that life, working, and management styles are significant predictors of work stress, and moreover that management styles are significant predictors of work stress above the life and working styles. The management styles are observed from the point of view of the theoretical models by Hersey and Blanchard, while stress is observed from the transactional theory of stress perspective. The research study was carried out on a sample of 618 subjects of both sexes (344 female subjects and 274 respondents are male), aged 19 to 59 years, with different educational backgrounds, and in various managerial positions in operations and administration. The following questionnaires were used; the Kenneth–Blanchard test of situational leadership, the Four-Dimensional Symptom Questionnaire for determining stress levels, the modified Allport–Vernon–Lindsay Scale of Values, and Julie Hay's Working Styles Questionnaire. The results confirm that the family–sentimental, utilitarian, and knowledge styles are the most significant value orientations for a manager. More than half of the managers have an increase of the indicators of work stress, wherein distress and somatization are more pronounced than depression and anxiety. Greater satisfaction with the balance of private and professional area life is related to less pronounced symptoms of work stress and vice versa.

The results of the survey show the most important value orientation of managers is whether they are under stress and how they react to it. The structure of the prediction of the work stress of the manager was examined based on their preference of management style, that is, working and lifestyle. The results indicate that all dimensions of stress at work can be explained by all three styles. The relationship between management style and employee satisfaction was also explored. The results of this research indicate that the style of management is best when it is coordinated with employees [5].

This research provides insight into the contribution of management styles to the balance of the private and professional life segments, as well as stress reduction in managers. The findings, based on a sample of managers of small businesses and private entrepreneurship, triggered additional questions about the direction of the influence of other internal and external factors of coping with work stress, a manager's capacity to overcome stress, and the most common strategies used for overcoming work stress, as well as the link between preferences of life and working styles and management styles. The research conducted also has practical implications. The results obtained give suggestions intended to increase satisfaction through the balancing of life and work among managers. Meeting practical needs implies the development of awareness and care about the psychosomatic health of employees and their professional development. The managerial implication of the research is the implementation of sustainable human resource development strategies and development programs [6].

The psychology of sustainability and sustainable development uses new aspects to find more effective strategies and solutions in work through professional and individual development [7]. The psychology of sustainability and sustainable development strives to contribute to the sustainability and sustainable development of every person [8]. Healthy organizations can be assessed not only from a financial point of view, but also from a humanistic point of view [9]. The organization's competitiveness is proportional to the experience, knowledge, skills, and competences of its employees [10]. A high

level of flexibility and the organization's readiness for understanding are necessary for defining work dynamics and employee health. In the work environment, employees respond to stressful events with their personal and professional working styles. Employees subjectively evaluate the stress using a transactional stress model. The reaction and work of employees is related to their assessment of whether they work in a healthy or unhealthy environment [11].

The development of an organization can focus on the development of certain professional competences and the work styles of employees, but it can also focus on personal development and self-perceived employability in order to identify the causes of stress and distress, and expand the capacities for a functional response [12]. The logic of the work–life balance concept, businesswise, is based on productivity which is a result of a person's higher motivation and dedication when personal responsibilities are considered. It is important to create a culture that supports the balance between business and private life and thus promote sustainable human development through a win-win strategy [13]. Also, the psychology of sustainability and sustainable development can be seen therefore as a new access point for well-being in organizations [14].

## 2. Theoretical Background

### 2.1. Working Style

The preference of working styles and management styles is a part of organizational culture and they are particularly significant since managers represent a model of identification for other employees [15,16]. The theoretical framework for understanding working styles in this study is the forced behavior taken from the theoretical model of transactional analysis, by working out the concept in which drivers are seen as the initiators of action in the work environment. Script imperatives (behavior drivers) rely on injunctions. Transactional analysis is successfully applied in the clinical, educational, organizational, and advisory fields of specialization. It is based on the study of interactions among people, and as the name implies, the analysis of interpersonal transactions. Its creator is Erik Bern, a Canadian psychiatrist who created Transactional Analysis as an introduction to psychoanalysis, with the intention of making it more concise and efficient and more comprehensible and adapted to most psychiatric patients. Teibi Keller classifies parent messages into five typical drivers: "Be perfect", "Be strong", "Please others", "Hurry up", and "Try hard".

A person with the driver "Be perfect" strives for perfection in all aspects of life, that is, in whatever is especially important to their (work, appearance, hygiene, knowledge, love, etc.) or in several of them. Such a person spends a lot of energy and time (or even all of it) on those aspects of life in which they want to achieve "perfection", so that there is not enough time for other aspects of life. The pursuit of perfection can be comparative (they must be better than others), or it is incompatible, when a person seeks to "overcome themselves" to do things the best they can. Perfectionists are unproductive because they irrationally spend a lot of time on achieving unnecessarily good results.

People with the "Be strong" driver—this driver implies working well under pressure and working well in crisis situations—can take control so that other people feel safe in their environment. When others in their vicinity are prone to "panic" reactions, these people usually think very logically, they can be emotionally distanced from the situation; on the other hand, they often do not recognize personal weaknesses, they have a deficiency of emotions (they can seem to be robots).

"Please others"—A person who satisfies others and fulfills their wishes even before others ask for it usually has no wishes, is unable to say "no", does not ask for what they need, does not show anger, does not set boundaries, eliminates conflicts (does not refuse, does not demand, does not require), and feels responsibility for the feelings of other people. A person with this driver loses social power, retreats from communication, and remains separate from their own wishes. They are good team players, empathetic, prefer to love others, avoid conflicts, provide constructive criticism, save others, and do not express their opinions; these qualities can frustrate people in their environment.

"Hurry up"—This professional style is based on the idea of productivity and efficiency (the faster, the more). A person with this driver is accelerated in everything they do and hardly tolerate social and other situations in which the activities take place at a normal rhythm or more slowly. These are people who download too many things at the same time, are constantly in a hurry, on the verge of patience, and often do not pay attention to details.

"Try hard"—implies an imperative—you have to work hard, without reaching the goals set. People with this driver have a tense and painful facial expression, often asking multiple questions at the same time without waiting for an answer and are inclined to say that most things are impossible. They have a problem of closure, and are afraid of mistakes and responsibility. These are people who can do great pioneering jobs, love new projects, and can start a lot of new things under stress, but often have a problem because they run a lot of jobs and activities that do not come to an end, they are more focused on "trying" than on success. Hay [17] offers guidelines to the contribution of behavior drivers (Transactional Analysis) and describes understanding organizational behavior based on these imperatives and how they contribute to organizational culture. There is some evidence of a direct link between managerial style and employee well-being [18]. The psychology of sustainability and sustainable development includes managerial styles that recognize the importance of relationships in organizational contexts for the well-being of workers. On a sample of 204 students from the Faculty of Philosophy in Serbia, research showed that the two most impelling motivation drivers are Be strong and Please others, followed by the Work hard motivational driver [19]. Research confirms that certain professional styles result in highly productive employees, but they can be an obstacle in achieving a balance between life and work. Obstacles are often manifested through stress at work [20].

## 2.2. Lifestyle

The term lifestyle is determined by the ways leisure time is spent and needs are met, the ways of spending resources, and the characteristic interpersonal and social relationships established by an individual. It is also about physical, mental and social well-being. A health-oriented lifestyle contributes to a healthy attitude towards work [21]. Values direct managerial actions towards the goals which they consider desirable and they have a motivational effect on behavior [22]. Lifestyle and a stressful life have an influence on the health and work of employees [23].

Values set requirements relative to the mind and activity of an individual and thus have a certain authority over them [24]. Because values require dedication to certain goals and behaviors, they shape a person's life.

There are several guidelines in the field of value research. The theoretical framework used in this paper is the view of All port which sees values as beliefs that are in line with human actions, giving them a special meaning of some kind of spiritual motivation factor in human behavior. According to his theory, every mature person has a built-in value system that gives meaning to his life. Different people have different core values that vigorously and constantly move them to do certain activities. Starting from philosophical and anthropological considerations, All port distinguishes the following six basic values: Theoretical, Economic, Aesthetic, Social, Political, and Religious. On the basis of the above typology, Allport, Vernon, and Lindsay developed a scale for measuring the intensity of lifestyle preferences [25]. Later, the authors introduced changes in the conceptualization and method of measurement in their investigations. In several studies, these descriptions of lifestyles were examined on a sample of students of the eight grades of elementary school, where the metric characteristics of the instrument were tested [26].

In this paper, managers' lifestyle preferences will be tested by a modified Allport–Vernon–Lindsay scale of values that measures the intensity of the preference of ten lifestyles on a five-step scale: utilitarian style, family–sentimental style, egoistic orientation, orientation to popularity, hedonistic orientation, orientation toward power, Prometheus activism, altruistic orientation, cognitive style, and religious–traditional style. The following are descriptions of the stated lifestyles:

1. Family–sentimental—Meet the person you love and who loves you, be together with them, establish a family, and dedicate yourself to them completely. Find the meaning of life in the family.
2. Hedonistic—Live in the present and enjoy pleasure—because the future is quite uncertain, and life is transient. Earned money does not need to be saved, but consumed to achieve as much immediate satisfaction in life as possible.
3. Utilitarian—Doing a well-paid job that provides good earnings and total financial security. Provide a rich and comfortable life for yourself and your family.
4. Altruistic—To do something useful for other people, to help them when they are unhappy or endangered, to be gracious and generous, even at the cost of personal sacrifice.
5. Egoistic—Approach life so that you do not depend on others and you do not have to worry about someone else's worries. First of all, be concerned about yourself and your well-being.
6. Religious–traditional—Believe in God and live in harmony with the teaching of your faith. In religion, try to find peace and truth about life. To be a good believer, respect religious holidays and religious customs.
7. Orientation to popularity—Become popular, be famous for sports, music, or art. Often appear in public, have a lot of fans.
8. Orientation to Power—Choose a life that provides great power, reputation and respect in society. Have a significant and recognized place and have a big impact on other people.
9. Knowledge (knowledge orientation)—Do research, search for new inventions and discoveries. To have as much knowledge as possible. Attend to seeking truth and studying nature, society and man.
10. Prometheus activism—Firmly strive to create a better and fairer relationship in the environment and society. Fight for distant goals and ideas, even when we do not succeed and when we encounter resistance in the environment. When it is necessary to give up immediate satisfaction at the expense of these ideas.

The research of the change in value orientations in the last two decades in our country indicates a decline in the popularity of lifestyles that implies advocacy for the general interests and well-being of other people, while lifestyles that focus on personal well-being predominate [27].

The value orientations of managers as well as their preferred lifestyles on the one hand are a product of the general state of society, and on the other hand, the product of the main actors in the organization [28].

*2.3. Management Style*

The theoretical framework for understanding the management styles in this paper is the Hersey–Blanchard Situational Leadership Model [29], and it represents one of the most contingent models, because it is dynamic and flexible. It starts from the concept of a continuum, with the leadership being aimed at a task on one hand, and people on the other [30,31].Management styles influence managerial behavior towards a qualitative approach [32].The situational modeling of leadership developed by Paul Hersey and Kenneth–Blanchard is one of the most important contingent models according to which the most effective leadership style changes depending on the "readiness" of the workers, which the authors define as a desire to prove, a willingness to accept responsibility as well as skills, and have the skills and experience needed to perform tasks. Therefore, the goals and knowledge of the followers are important elements in the process of determining an effective leadership style.

Hersey and Blanchard believe that relationships between managers and followers go through four developmental stages, and that the manager should apply a leadership style in accordance with the degree of development of his followers. In the initial phase, the most appropriate is the orientation of the managers to the task. Collaborators need to get a solid structure, instructions on tasks, rules and procedures. A manager who is not authoritarian in his attitude can cause concern and confusion in new followers. In the second phase of learning, behavior-oriented action remains the basic model,

since followers are still not ready to function without a structure. The trust of leaders in associates and their support are becoming more and more familiar to them, so that they increasingly focus on employee relationships. In the third phase, associates become more capable and motivated to prove themselves, and they are increasingly actively seeking greater responsibility. Therefore, the leader does not have to be authoritarian anymore and should provide support to associates, be careful and support the followers' determination to take on greater responsibility. In the final phase, the followers no longer need guidance from managers, because they are more experienced, more confident and more independent. Situation theories describe the way in which the situation shapes the relationship between the conduct of the leader and the results, and suggests that effective leadership requires a rational understanding of the situation and an appropriate response, more than a charismatic leader with a large group of devoted followers [31].

In accordance with the theoretical model of Hersey and Blanchard, this paper starts from the basic concept of the existence of two dimensions of focus that the manager uses in their work: focus on the task (goal) or focus on people. In this paper we take into account only the individual aspect, since it was extremely complicated to set up a model in which we could evaluate the behavior of managers as seen by the team that it is guiding.

After processing the response, the results can be grouped into one of the following categories.

1.  Directing—The manager provides detailed instructions to the associates and very closely guides associates.
2.  Telling—When delegating, the manager defines principles, gives reasoning, and engages collaborators more.
3.  Participating—The manager assists colleagues in terms of clarification, leads when necessary.
4.  Delegating—The goal is clear, and the job is left to collaborators.

*2.4. Stress in Organization*

Numerous studies on work stress, which is related to the difficulty and content of work, dissatisfaction, and not being adjusted at work, confirm different effects of work stress [33–35]. Whether it is due to the excessive demands of the workplace, job insecurity, irregular salaries or tight deadlines, work is often a great source of stress for a modern individual [36].

Professional stress involves a whole range of harmful physiological, psychological and behavioral reactions to situations [37]. An individual with a lower stress level will be more productive, more pleased and more motivated to work as numerous studies have indicated [38]. Siegrist [39] indicates that the model effort–reward imbalance is used in predicting health and in strengthening work-related relationships.

The research in this paper is based on the model of organizational health by Hart and Cooper based on Lazarus' transactional model of stress [40] and Hart's model [41], which takes into account the well-being of an individual and an organization, i.e., its financial goals and social responsibility. Sources of work stress (stressors) can be individual—arising from the work role; group—caused by the group's dynamics and managers' behavior; or organizational—arising from the organization's characteristics. Mobbing is a commonly studied and proven source of stress [42].

Stressors related to career/job insecurity affect self-esteem [43], and they can lead to serious health complications: organizational factors—organizational structure, culture, changes, communication, and performance [44]. Stress is a partial mediator in the relationship between the challenges in the work-home relation and job satisfaction [45]. The work–family conflict is related to the overall well-being of the employees and the organization.

## 3. The Present Study

With the given framework as its basis, this study addresses the conceptual and theoretical issues relating to the linkage of leadership styles, the styles of living of professional managers, and stress at

work. The goal is to study the relationships between stress styles, as well as determine the prediction of stress based on styles.

There are individual studies of the association between life, working and management styles, and stress, but as far as we know none of the studies has offered an explanation of this association.

A key indicator of a healthy organization is the way employees do their jobs, see their jobs, and perform their work qualitatively with as little stress as possible. Thus, we explored the prediction of work stress based on life, working, and management styles. Furthermore, we explore the contribution of working and management styles above and beyond the lifestyles in order to get better insight into the importance of these professional styles in work stress prediction. Thus, we expected that management styles contribute to the prediction of work stress beyond life and working styles.

## 4. Method

### 4.1. Participants

Respecting the rules on forming sample size ensures representativeness or the possibility of obtaining a valid conclusion from the sample to the overall population. The sample in this study is suitable and it consists of 618 managers, of varying hierarchical levels, employed in a service-type enterprise from the entire territory of the Republic of Serbia. The sample is comprised of managers of both sexes, different hierarchical positions in the organization and the various types of work they perform in order to represent the population as faithfully as possible. Operational managers are involved, that is, in retail and wholesale facilities of different sizes (sales managers, i.e., managers and their deputies, and heads of departments in large sales units), as well as in executive services (executive directors, sector managers, heads of departments, or managers of departments).

The sample included managers of a service company. The study was conducted in 2018 on a sample of 618 respondents of both sexes (344 women and 274 men), aged 19 to 59; 76% of whom are married and have one or more children. The average length of the work path is 16 years, while the average length of the workplace in the company in which the research was conducted is 7 years among respondents. In accordance with the hierarchical structure of the organization, the largest proportion of respondents in the sample belongs to the category of lower management, 80.9%; the middle management category accounts for 15.7% of respondents; and the senior management category accounts for 3.4%. More than three-quarters of the respondents (86.08%) are managers in operations, that is, retail and wholesale facilities, while 86 (13.92%) manage associates in professional services.

A total of 940 managers work at managerial positions in the company at different hierarchical levels, of which 110 in professional services, while 830 in operations.

Data was collected by individual and group testing, with instruments belonging to the class of group tests. The test was anonymous and carried out on a voluntary basis. It only took about half an hour to complete the questionnaire. Respondents were given very detailed instructions.

The Ethics Committee made up of the Managing Director together with the Board of Directors approved this research and provided key feedback on the research. The feedback has had a practical benefit in order to improve business and solve key problems so that managers are more satisfied.

### 4.2. Measures

A behavior driver test by Julie Hay (1997) identified the preference of working styles, which are typical also in the framework of work behavior: an integral part of the transactional analysis model: "Be perfect", "Be strong", "Please others", "Hurry up", and "Try hard". The questionnaire comprised 25 items and each answer was scored 0 to 8, depending on the degree of relation between the described behavior and the respondent. The result is information about the intensity of each of the following drivers as professional styles. Examples of items from the questionnaire: I finish working tasks faster than other people, Sometimes I have a problem saying "no" to others, although I already have too many obligations, I have the custom of waiting for the last moment and then just start the task (job),

and I am much more enthusiastic than others. This query was made by researchers in the study of employee behavior [46–48].

To determine the preference of management styles, the Kenneth–Blanchard test of situational leadership was used for the 12 given situations. A description of one situation from the questionnaire—Your group members are not able to solve the problem themselves (the respondent circles one of the four responses), the group usually works independently, team performance and mutual relationships are good, etc. This paper starts from the basic concept of the existence of two dimensions of focus that the manager has in his work: focus on the task (goal), or focus on people. Participants choose one response that is most closely related to their reaction in a given situation, and from the angle of the manager as an individual, that is, from the angle of the group they lead.

The results are classified into four categories:

- Directing—the manager gives detailed instructions and closely leads the coworkers.
- Retelling—while delegating, one gives explanations and principles, engages the coworkers to a greater extent.
- Participating—assists the coworkers in terms of explanations, leads when necessary.
- Delegating—the goal is clear, and work is left to coworkers.

Many researchers have been studying managerial styles and contributing to theory and practice [49,50]. By using the modified Allport–Vernon–Lindsay Scale of Values, preferences of ten lifestyles have been identified: utilitarian, family–sentimental, knowledge and religious–traditional style, Promethean activism, egoistic orientation, popularity-oriented, hedonistic orientation, power orientation, altruistic orientation [51]. These authors created the scale based on the typology of values, which includes theoretical, economic, aesthetic, social, religious, and political value orientations.

The questionnaire offers short descriptions for the above 10 lifestyles, and each of them responds to a five-step scale to what extent they would like to live in the described way (the maximum value is 5—"I like this way of life very much"—and the minimum is 1—"I do not like this way of life at all"). In the end, they explain which of the offered styles is the most, or at least, liked, and the style of life most suited to the one they currently live in. This scale was validated in a large number of studies when the metric characteristics of the instrument were also checked [40,51]. In this research, he will take over the lifestyles that Popadić used in his research "The Lifetime and Generation Differences in Lifetime Preference", which the author claims to be concise and simple.

Many other authors used this questionnaire in research [52–54].

Stress level was measured by using the Four-Dimensional Symptom Questionnaire (4DSQ), intended for the nonclinical population, which makes a distinction between a general feeling of distress and the occurrence of psychopathological symptoms. The questionnaire is made up of four scales which include 50 items: distress, depression, anxiety, and somatization. The dimension of the distress relates to the symptoms of stress, which result from the action of the stressors and the efforts made to minimize them [55,56]. Depression refers to the existence of depressive thoughts, including suicidal ideas and the loss of a sense of satisfaction (anhedonia), which presents symptoms of mood disorders. Anxiety refers to the existence of symptoms of free-floating anxiety, panic attacks, phobias, and avoiding behaviors characterized by anxiety disorders. The dimension of somatization is psychosomatic symptoms (pain in the muscles, neck, back, headaches, stomach problems, heart palpitations, and lack of breath). The metric characteristics of the 4DSQ instrument were examined on a sample of employees in Dutch Telekom: Alpha coefficients are distress 0.90, depression 0.82, anxiety 0.79, and somatization 0.80. Since all alpha coefficients are greater than 0.70, it is considered that the questionnaire has good internal consistency. The response is gathered by summing up points from individual points within each scale, whereby the answer never gives 0 points, sometimes 1 point, while the answer is often 2 points.

## 5. Results

First, correlations between life, working and management styles with four aspects of work stress were calculated (Table 1), and then predictions of each work stress aspect based on life, working and management style were calculated by hierarchical regression analysis.

**Table 1.** Correlation of life, working, and management styles with work stress.

|  | Distress | Depression | Anxiety | Somatization |
|---|---|---|---|---|
| Family–sentimental | −0.06 | −0.03 | −0.03 | −0.06 |
| Altruistic-oriented | 0.01 | 0.01 | −0.02 | 0.05 |
| Knowledge-oriented | −0.09 * | −0.05 | −0.07 | −0.07 |
| Utilitarian | −0.01 | −0.01 | 0.05 | 0.00 |
| Popularity-oriented | −0.05 | −0.01 | 0.06 | −0.08 |
| Egoistic | 0.09 * | 0.11 ** | 0.08 * | 0.08 * |
| Promethean activism | −0.08 | −0.03 | −0.09 * | −0.08 * |
| Hedonistic | 0.07 | 0.09 * | 0.06 | 0.01 |
| Religious–traditional | −0.03 | −0.04 | 0.05 | 0.02 |
| Power-oriented | −0.070 | −0.04 | −0.05 | −0.08 |
| Hurry up | 0.25 ** | 0.23 ** | 0.15 ** | 0.22 ** |
| Be perfect | −0.08 | −0.04 | −0.05 | −0.06 |
| Please others | 0.16 ** | 0.15 ** | 0.17 ** | 0.17 ** |
| Try hard | −0.01 | 0.01 | −0.00 | 0.03 |
| Be strong | −0.03 | −0.01 | −0.03 | 0.02 |
| Directing | −0.14 ** | −0.08 | −0.06 | −0.12 ** |
| Telling | 0.03 | −0.05 | −0.03 | 0.06 |
| Participating | 0.12 ** | 0.13 ** | 0.09 * | 0.08 * |
| Delegating | 0.04 | 0.07 | 0.04 | −0.02 |

** $p < 0.01$, * $p < 0.05$.

Of all lifestyles, the Egoistic lifestyle showed significant correlation with all dimensions of work stress. The Knowledge-oriented lifestyle showed significant correlation with distress, and Promethean activism showed significant negative correlation with anxiety and somatization. As for the working style, Hurry up and Please others showed a significant positive correlation with all aspects of work stress. As regards the management styles, the Participating style showed a significant positive correlation with all aspects of work stress, and the Directing style showed a negative correlation with distress and somatization (Table 1).

The results of the descriptive data and reliability for life, working, and management styles confirm that the subjects are generally characterized by family–sentimental (Mean = 4.46 SD = 0.77) and utilitarian style (Mean = 4.20 SD = 0.65). Most highly marked working styles were "Be perfect" (Mean = 27.62 SD = 4.96) and "Please others" (Mean = 27.68 SD = 5.35), and among management styles, the managers mostly prefer Telling (Mean = 5.38 SD = 1.82)

The results of a hierarchical regression analysis (Table 2) showed that management styles contribute to the prediction of distress and depression, above and beyond the life and working styles. It could be noted that working styles contribute more to stress prediction than lifestyles.

**Table 2.** Hierarchical regression analysis: prediction of stress based on life, working and management styles.

| Blocks | Predictors | Distress | | Depression | | Anxiety | | Somatization | |
|---|---|---|---|---|---|---|---|---|---|
| | | β | r | β | r | β | r | β | r |
| Lifestyles | Family–sentimental | −0.063 | −0.075 | −0.032 | −0.041 | −0.065 | −0.039 | −0.087 * | −0.070 |
| | Altruistic-oriented | 0.017 | 0.005 | −0.011 | −0.002 | −0.037 | −0.013 | 0.053 | 0.042 |
| | Knowledge-oriented | −0.058 | −0.094 | −0.034 | −0.057 | −0.024 | −0.059 | −0.050 + | −0.080 |
| | Utilitarian | 0.055 | 0.004 | 0.018 | 0.003 | 0.078 + | 0.057 | 0.080 | 0.013 |
| | Popularity-oriented | −0.028 | −0.057 | −0.004 | −0.015 | 0.093 * | 0.056 | −0.063 | −0.080 |
| | Egoistic | 0.076 + | 0.088 | 0.086 * | 0.108 | 0.061 | 0.084 | 0.083 * | 0.078 |
| | Promethean activism | −0.054 | −0.069 | −0.027 | −0.026 | −0.111 * | −0.075 | −0.084 * | −0.081 |
| | Hedonistic | 0.027 | 0.066 | 0.046 | 0.089 | 0.013 | 0.063 | −0.046 | −0.003 |
| | Religious–traditional | 0.005 | −0.029 | −0.022 | −0.031 | 0.083 + | 0.051 | 0.069 | 0.022 |
| | Power-oriented | −0.044 | −0.073 | −0.032 | −0.040 | −0.088 + | −0.049 | −0.040 | −0.079 |
| | R2 | 0.042 ** | | 0.031 * | | 0.041 ** | | 0.045 ** | |
| Working styles | Hurry up | 0.245 ** | 0.266 | 0.208 ** | 0.240 | 0.132 ** | 0.163 | 0.206 ** | 0.238 |
| | Be perfect | −0.061 | −0.072 | −0.032 | −0.041 | −0.034 | −0.035 | −0.079 + | −0.057 |
| | Please others | 123 ** | 0.152 | 0.138 ** | 0.159 | 0.171 ** | 0.159 | 0.121 ** | 0.166 |
| | Try hard | −0.033 | −0.008 | −0.036 | 0.005 | −0.011 | −0.004 | 0.016 | 0.039 |
| | Be strong | −0.079 + | −0.029 | −0.072 | −0.014 | −0.109 ** | −0.039 | −0.024 | 0.028 |
| | ΔR2 | 0.082 ** | | 0.064 ** | | 0.051 ** | | 0.069 ** | |
| Management styles | Directing | −0.059 | −0.133 | 0.024 | −0.073 | 0.004 | −0.047 | −0.066 | −0.116 |
| | Delegating | 0.041 | 0.070 | 0.074 + | 0.086 | 0.049 | 0.055 | −0.012 | 0.006 |
| | Participating | 0.066 + | 0.108 | 0.132 ** | 0.130 | 0.057 | 0.062 | 0.036 | 0.080 |
| | ΔR2 | 0.013 ** | | 0.020 ** | | 0.005 | | 0.007 | |
| | Total R2 | 0.369 ** | | 0.339 ** | | 0.312 ** | | 0.347 ** | |

$** p < 0.01, * p < 0.05, + p < 0.07.$

As Table 2 shows, life and working styles are significant predictors of the all aspects of work stress, while the managerial styles significantly predict only distress and depression beyond life and working styles.

Analysis of partial predictor contributions has led to the conclusion that distress and depression can be explained based on the Lifestyle–Egoistic-oriented style.

The Working styles Hurry up and Please others are related to all aspects of work stress, where the Hurry up style was observed to have a slightly higher contribution regarding distress and the Please others style had a higher contribution regarding depression.

One of the management styles that has a significant influence on distress and depression is Participating. At first, in the domain of lifestyles, anxiety can be explained based on greater Popularity-oriented and lower Promethean activism. In the domain of professional styles, a significant prediction of anxiety is achieved by such styles as Hurry up, Please others, and Be Strong, the latter being used in a negative context. Somatization can be explained based on a more Egoistic orientation. In the domain of professional styles, somatization can be predicted based on stronger preferences for the following styles; Hurry up and Please people.

The results of the descriptive data and reliability for life, working and management styles confirm that the subjects are generally characterized by the family–sentimental (Mean = 4.46 SD = 0.77) and utilitarian style (Mean = 4.20 SD = 0.65). The most highly marked working styles were "Be perfect" (Mean = 27.62 SD = 4.96) and "Please others" (Mean = 27.68 SD = 5.35), and among management styles, the managers mostly prefer Telling (Mean = 5.38 SD = 1.82)

## 6. Discussion

The aim of the current study was to examine the prediction of work stress based on life, working and management styles, as well as the nature and direction of the mentioned relationships. Results showed that managerial styles predicted work stress beyond the contributions of life and work styles, which is

in line with our expectations. Hypothesis: We expected that life, working, and management styles are significant predictors of work stress, and moreover that management styles are significant predictors of work stress above the life and working styles. This was confirmed by the results of the regression analysis. The result accurately identifies the causes of the problem and can serve as a tool for preserving the mental health of employees.

Among management styles, distress and depression are only contributed to by Participating and, to some extent, Delegating, which contributes only to depression. Besides life and working styles, management styles have an effect on depression and distress to some extent. Management styles have an effect on depression, beside the variance of life and working styles, and to some extent on distress. With managers who often assist their collaborators in the process of clarification and directing them when necessary, we can anticipate depression. The assumption is that while managers are more socially investing in this exchange, they get more in return, and therefore the obtained finding requires a more detailed analysis. The research results show that the mutual trust between managers and employees and management style have a major effect on employees' efficiency [57].

Working styles contribute to predicting work stress more than lifestyles. The findings are understandable, because professional styles based on parental bans as drivers of behavior in the work environment reflect the insufficient adaptability of persons in different situations. Brief and colleagues [58] point out that some personality characteristics (perfectionism, distrust, rigid thinking, hostility, self-centeredness, and pessimism) are a prerequisite for more frequent manifestations of stress. Personality traits according to the results of previous studies have an important role in the development of stress. People who are more negative have traditionally been associated with greater distress, while sociable and positive people are generally psychologically healthier [59–62]. In the domain of working styles, the Please others and Hurry up styles have a significant prediction of anxiety and somatization. The individuals with the Please others style do not have social power, they withdraw from communication, they do not take care of their needs, failing to set the limits when necessary, avoid conflicts, and feel responsible for other people's feelings, which can be related to repression and psychosomatic difficulties. The working style Hurry up is found to be a predictor of anxiety, which we can understand in light of the fact that people with a professional style of preference "hurry up" are impatient, accelerated in everything they do, more difficult to handle in situations where activities take place in a normal rhythm or even more slowly, and are based on the idea of productivity and efficiency; this probably arises from a basic feeling of insecurity and anxiety to adequately carry out tasks, which can certainly be linked to anxiety. It would be interesting to further analyze this segment. The findings can be related to the previous theoretical findings and most research findings suggesting a relationship between certain dimensions of stress and anxiety [63–65].

In the lifestyles domain, the Egoistic orientation and Promethean activism contribute most to predicting different indicators of work stress, with the share of working styles being bigger (especially Hurry up and Please others). Managers dominated by the Egoist orientation do not depend on others, do not worry about others' concerns, they take care of themselves and their well-being, they are more inclined to somatizing. This can be seen as a form of focus on one's own being, that is, closure and suppression of emotions, which ultimately leads to the more frequent manifestation of psychosomatic symptoms. Promethean activism in the behavior of managers is reflected through persistent efforts to create better and fairer relationships in the environment and society. They are struggling for goals and ideas, even when they do not succeed and when they encounter resistance in the environment. Anxiety can be explained based on a higher orientation on popularity and lower Promethean activism.

The contributions of the religious–traditional and utilitarian styles to anxiety are marginal, but the impact is positive, while the orientation on power makes a negative marginal connection with anxiety. It is understandable that religious people are more modest and humble, and that they express anxiety to a greater extent, while those who want power, prestige, and respect in society show a lower level of anxiety, which confirms the connection between burnout and anxiety.

The results obtained can be related to the previous theoretical knowledge and most research findings which point to the existence of a relation between specific dimensions of stressful experience and anxiety [66].

## 7. Conclusions

Individuals are facing new challenges for career management and life management arising from the complexity of the current world of work. The goal of an organization is to achieve business targets and intensive development, as well as focus on the interests of its key employees, the achievement of their personal and professional goals, the harmonization of life and work. Organizations can help employees by introducing them to a new concept called Positive Lifelong Management, which contributes to seeing personal and organizational goals together [67]. The company in which this survey has been conducted has a very clear and rigid hierarchical structure, with an organizational culture in which very high demands are communicated to managers on all levels very openly. The priority is that the values and competencies of the organization are transparent and that the value system of a manager is similar to the organization's value system.

The results of this paper should serve to correct and modify styles, especially managerial and working styles. The stated hypothesis was confirmed based on the regression analysis, indicating the prediction of stress on the various styles mentioned. Based on this, it is necessary to create a support environment that allows the members to slow down and set realistic tasks. It is necessary for the top management to have more understanding and to direct the business equally toward tasks and toward people. In a healthier environment, people will have the freedom to need as well as to get permission to establish a balance between life and work. Such implemented measures will directly reduce stress. The logic of the life balance is that people are more motivated and more productive because they have control over their time and work. However, findings indicate that managers prefer working styles that provide them a better place in the work environment and compliance with the values on which the organizational culture rests. According to importance, the most important is the management style, then the working style and the end-of-life style. Management styles mostly affect the occurrence of stress and anxiety. The management style that has a statistically significant impact on distress and depression is Participating. The dimension of distress refers to the symptoms of stress, which arise as a result of the actions of the stressor and the efforts made to minimize them, and Depression refers to the existence of depressive thoughts, including suicidal ideas and the loss of a sense of satisfaction that presents symptoms of mood disorders. It is necessary to analyze the needs of employees as well as their new autonomy and work engagement in order to determine the most appropriate way of guiding them. Anxiety can be explained based on more Popularity-oriented and lower Promethean activism.

In the domain of working styles, a significant role in predicting anxiety and somatization in a positive context is assumed by the Please others and Hurry up styles. In accordance with Lazarus & Folkman's Transactional Theory of Lazarus [68], which emphasizes that the pressure of a stressful experience is a special relationship between a person and their environment, it is important to emphasize that the organization in which this test was conducted has a very clear and rigid hierarchical structure, with an organizational culture in which very high demands are communicated very openly to managers at all levels. In this organization, managers prefer the style of Hurry up, which refers to taking a large number of tasks onto themselves, speeding up the work, and short deadlines, which fits into an organizational culture in which the aspiration to excellence is extremely highly valued. Another preferred style is a professional style—Attend to others—which testifies to the efforts of managers to adapt to the expectations of the highest leadership of the organization; it is demanding and consistent in implementing the given way of managing the organization, and as a measure of loyalty and support to the organization, belief in its values and goals, and the willingness to invest the expected effort. Somatization in the domain of personal style can be explained based on a more Egoistic orientation and on Promethean activism. Somatization as a form of stress can be explained by amore egoistic orientation in part of the lifestyle. Aspects of stress in work distress and

somatization are more pronounced in relation to depression and anxiety. The estimation of stress and the examination of given styles can serve as a basis for the development of adequate anti-stress programs and measures that would contribute to better psychological selection, work adaptation, better career management, and the organization of work.

The results unambiguously indicate what needs to be changed in the style of management as well as in the working styles of employees. It is necessary to make a small step towards a work–life balance that would provide insights into the opening of the creative process and options in decision-making, self-discipline, and the desire to successfully deal with changes.

The theoretical contribution of this paper is reflected in the provision of additional information aimed at examining the relationships between the preferences of certain styles of management, working style and lifestyle, and the exposure to the thinking of the manager, as well as a better understanding of concepts in particular. The practical implication is to increase the satisfaction of the work life balance with managers, determine their needs and expectations, identify the most effective management styles in the organization that will contribute to the balance of professional and lifestyle management and the development of the management process, creating a realistic view of employee careers.

The practical contribution of this study is reflected in the use of professional and lifestyles in the process of professional selection, as well as the integration of the value orientation of employees and management with the aim of fostering organizational culture.

The findings of research triggered additional questions about the direction of the influence of other internal and external factors of coping with work stress. The research provides insight into the contribution of management styles to the balance of the private and professional life areas, as well as stress reduction in managers [69]. It is recommended that the management of stress in the organization be recognized as a valuable investment for employees, because the importance of stress for both the individual and the organization is enormous. It is important from the aspect of social policy and society in general, because raising awareness to the level of significance of the individual and the organization makes up the heart of society. The implementation of an employee health care program, as well as the establishment of a trust-based working time system [70] would contribute to the performance of teams and individuals, as well as the whole organization. The four highest ranked measures at the level of the organization for improving the health of the employees are time to check health status, changes in the ergonomic conditions in the work environment, redefining the work of employees, and conducting training and education in the field of a healthy lifestyle [71]. The findings of a UK study confirm that quality leadership implies social support and an adequate job design, which is a predictor of employee satisfaction and loyalty [72]. For people under labor stress, it is very important that they learn to recognize it and more effectively cope with it in order to preserve their mental health and provide more efficient work.

The findings of the research in this paper indicate the risks arising from certain working styles and their relation to stress at work.

The results of the research confirm that careful management implies social support and adequate job design which are predictors of job satisfaction, employee loyalty and reduced work-related stress. Giorgi et al. [73] show that workplace stress has negative effects on workers' and organizations' psychological and physical health. It is of great importance for people suffering from work-related stress to learn how to recognize it and easily cope with it for the purpose of maintaining mental health and efficient work. On the basis of the results obtained, interventions can be created that integrate personal counseling and career counseling, promoting the development of sustainable career projects in creating such a psychological climate in which care for people in each organization becomes one of the key values of the business [74,75]. Our research highlights the significance of healthy and safe environments and the promotion of well-being and the quality of life of individuals within an organization. The implications of the research are useful for the organization in order to improve human resources and management policies and practices [76,77].

*Limitations and Future Directions*

The limitation of the current study includes location because the study was conducted in Serbia and the results were related to this area, although the study was innovative for Serbia because the topic was investigated for the first time in this manner. The basic limitation of the research relates to the type of design of the research itself. On the basis of the data obtained, it is possible to conclude with certainty only the existence of significant relationships among the examined variables, while with caution, it can be assumed that the cause–effect relationships between the examined constructs exist as long as they are not checked by a longitudinal design. Consequently, one of the implications for future research is the monitoring of the proposed predictors of the stress model at work. It is an implication for future research to create an experimental or longitudinal design. It would be interesting to design a model of the research that would enable the comparison of the examined variables between the organizations in developing countries as well as a comparison between countries in transition and developed countries. The study shows the relationship only between the examined variables; however, it is possible to analyze the variables of a manager's capacity to handle work-related stress and his/her most common strategies for that. The future research in this field can be focused on the analysis of internal and external factors in coping with work-related stress, then on the connection between preferences for life and working styles and management styles on the sample of managers from numerous industries in which organizational cultures are significantly different, as well as their attitude to job responsibilities and approach to business processes management. It would be interesting to check the situation regarding the stated goals in other organizations (smaller organizations), other economic branches (production organizations), and then in state institutions, as well as in the field of private entrepreneurship, in which there are significantly different characteristics of organizational cultures, attitudes towards business obligations, and approaches to managing associates and managing business processes. Another important challenge will be focusing on the key psychological aspects of managers towards their well-being and sustainable development.

**Author Contributions:** Conceptualization was made by I.K., T.K. and A.I.; Methodology was developed by I.K. and M.L.; Investigation was made by N.B. and A.I. Writing—original draft preparation, I.K., T.K. and A.I.; Writing—review and editing was made by N.B. and M.L.

**Funding:** This research received no external funding.

**Conflicts of Interest:** The authors declare no conflict of interest.

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
