# Peer review of "The Impact of Stress on Life, Working, and Management Styles: How to Make an Organization Healthier?"

_sustainability, doi:10.3390/su11154026_

Round 1
Reviewer 1 Report
In general, this is an interesting research, but the concept of the paper needs to be improved, with theoretical and empirical part more connected.
Make an additional grammar check of your paper. For instance in your abstract you have a verb or object missing in the following sentence: „a balance between life and work in order to of the psychology of sustainability“; afterwards you have out but it should be our etc.
Consider simplifying your sentences. Sometimes they are hard to follow and understand – „For example – „In work environment, employees respond with their personal and professional working styles to a stressful event, which according to the transactional model of stress involves subjective evaluation of employees- does the organization promote healthy work environment? „
Please divide introduction from the theoretical part of your paper. The introduction should state what we know, what we do not know (research gaps and study motivation) and how our study is contributing. Suggestion- Form the introduction in the following manner: 1st paragraph - the current knowledge on the topic 2nd paragraph - direction toward the purpose of the paper 3rd paragraph - the purpose of the paper and it states briefly methodology that has been utilized in the paper 4th paragraph – explain what is the contribution of the paper, in relation to several previous papers 5th paragraph - describe other sections of the paper.
Then have a theoretical part with literature review of relevant variables used in your study. As regard to current theoretical part (1.1. Life, working and management styles and Stress in organization) I find them inconclusive and “jumping” from one to another subject (especially part 1.1.). I suggest authors provide a wider picture on the subjects with reference to previous research in this field. Currently written I feel the paper misses a theoretical part with clear introduction into the problem and main drivers for their study. Currently it just provides insight into models they opted to use in their research – more appropriate under methodology part.
Describe how you selected your sample and some additional data about your sample.
In your discussion you say “hypothesis is confirmed“ but no hypothesis has been given previously, neither the background that led you to develop a certain hypotheses has been provided.
Provide additional theoretical and practical implications as well as research limitations.
Author Response
Dear reviewers,
Thank you for the constructive feedback you have provided regarding our manuscript. We are thankful for your comments, aimed to improve the publication of our research. We are glad that you find our topic interesting, and we hope that our revision will meet your criteria.
Below are a few comments about the changes that we have made in this version of the manuscript.
1. The biggest problem with this paper was the Introduction. We divided introduction from the theoretical part of paper. Introduction includes the current knowledge on the topic, the purpose of the paper , briefly methodology, contribution of the paper, describe other sections of the paper.
Therefore, we have changed the title of the manuscript. The title does adequately reflect the content of the paper.
2.We have made changes to the literature review, methodology section, results are clearly presented and the conclusion.
3. We corrected grammatical errors and and especially corrected English language and style
4.Additional literature was used, to better depict our perspective.
5. All the changes we made in the text are red.
Once again, we would like to thank you for your time spent in this review.
Page 1 - change the title, corrected grammatical error, corrected English language and style, new reference
Page 2 - change Introduction, divided in 5 part, corrected grammatical error, corrected English language and style
Page 3 – new references, new text, corrected grammatical error, corrected English language and style
Page 4 – added Theoretical background, more detailed style, revised part of the text, corrected grammatical error, corrected English language and style
Page 5 - revised part of the text, corrected grammatical error, corrected English language and style
Page 6 – added new text, corrected grammatical error, corrected English language and style
Page 7 - revised part of goals, hypothesis, better explained sample, corrected grammatical error, corrected English language and style
Page 8 - better explained Ethics Committee, better inform about their psychometric quality, corrected grammatical error, corrected English language and style
Page 9 - added some example of the items in questionnaire, more precisely explained results, corrected grammatical error, corrected English language and style
Page 10 - revised results, corrected grammatical error, corrected English language and style
Page 12 - better adjust conclusions to the findings, corrected grammatical error, corrected English language and style
Page 13,14,- added new text, Provide additional theoretical and practical implications, corrected grammatical error, corrected English language and style
Page 15 - revised limitations of research and future lines of investigations, corrected grammatical error, corrected English language and style
Page 16 - added 16 new references
Page 17 - added 8 new references
Page 18 - added 10 new references
Page 19 - added 8 new references

Reviewer 2 Report
Thank you for the opportunity to review this article. The time involved in submitting your manuscript is greatly appreciated.
Despite this, the article presents a series of issues that must be noted and mended. The recommendations are presented separately by sections. Hopefully, they would be useful.
Title: the title does not adequately reflect the content of the paper. Please, adapt it to better inform the readers about that content.
Introduction:
Firstly, some of the references that you cite are too old. Even though the most relevant studies should be referenced, also the RECENT research must be included. Moreover, I recommend a strong effort in applying the framework of Psychology of Sustainability, and Psychology of Sustainable Development as the theoretical umbrella that covers your research.
At the end of the literature review, the aims and the questions in the research should appear. Maybe to formulate the questions as a hypothesis would be an option to clear this aspect. Another commentary, it is the possibility of including this part at the final of the introduction part; even a separate section could be a good option, in order to clear the final of the introduction and to serve as a connection with the method.
Method:
Please, try to better describe the sociodemographic data of your participants. In the same sense, give the readers with detailed information about the procedure for recruiting participants and collecting data.
Which Ethical committee approved the study protocol? Please, explain it.
Related to the instruments, please better inform about their psychometric quality and give to the readers some example of the items. If you can, please inform about previous studies where the same instrument has been used and the reliability obtained in that research.
Data analyses
Please, explain to the readers which procedures of statistical analyses have been used and justify your decisions.
Results
The results should be presented following the same order as the introduction and hypotheses. Also, the same order must be used in the Tables. This simplifies the work for readers.
Discussion:
First of all, try to better adjust your conclusions to the findings. Or to say in other words, please try to justify more clearly the connection between your conclusions and your findings.
The most important comment, it is that some of the conclusions, related to the direct analysis of the results, should be revised.
Finally, a section related to limitations, future lines of investigations and the principal contributions of the research could be interesting. Your paper has a lot of relevant implications for society and policymakers, but you need to elaborate more on this topic.
Author Response

(The authors gave the same response as above.)

Round 2
Reviewer 1 Report
Dear authors, thank you for revising your paper and accepting our suggestions. In this form I find the paper acceptable, although I strongly encourage you to revise two things before publication. First, one additional English proof read is needed, as more fine tuning of English grammar and style is needed. Second, your mention in your introduction you have a hypothesis, but later on do not mention it, or reflect on it in your conclusion. So it would be good to revise this.
Author Response
Dear reviewer,
Thank you for the constructive feedback you have provided regarding our manuscript. We
are thankful for your comments, aimed to improve the publication of our research. We are
glad that you find our topic interesting, and we hope that our revision will meet your
criteria.
Below are a few comments about the changes that we have made in this version of the
manuscript.
1. We corrected grammatical errors and especially corrected English language and
style.
2. The hypothesis is mentioned in the part of the Discussion and the Conclusion section.
Once again, we would like to thank you for your time spent in this review.
Page 1 - corrected grammatical error, deleted - the, on,
Page 2 - corrected grammatical error, deleted - the, to, wants, A, corrected English
language and style
Page 3 – corrected grammatical error, deleted - order, coincides, its, e, tries, towards, to
stressful events,corrected English language and style
Page 4 – grammatical error, deleted- the, as, her, more, the, it, irrationally, with, the rest,
in, with, a, the, from, corrected English language and style
Page 5 - corrected grammatical error, delted- central, permanently, zi, e , renunciation,
for, corrected English language and style
Page 6 – corrected grammatical error, deleted - behavior, a, of behavior, as, corrected
English language and style
Page 7 - corrected grammatical error, deleted - the, it, corrected English language and style
Page 8 - corrected grammatical error,deleted- represented by, rounds off, corrected English language and style
Page 9 - corrected grammatical error, deleted- on, for, corrected English language and style
Page 12 - corrected grammatical error, deleted D, M, corrected English language and style
Page 13 - corrected grammatical error, deleted - slower, us , corrected English language and style
Page 14 - corrected grammatical error, deleted - of, impaired, which, the, corrected English language and style
Page 15 - corrected grammatical error, deleted- three, prevail, point to, significant, some, corrected English language and style
Page 16 - corrected grammatical error, deleted- the